# Herpes Virus Infection in Lung Transplantation: Diagnosis, Treatment and Prevention Strategies

**DOI:** 10.3390/v15122326

**Published:** 2023-11-27

**Authors:** Filippo Patrucco, Antonio Curtoni, Francesca Sidoti, Elisa Zanotto, Alessandro Bondi, Carlo Albera, Massimo Boffini, Rossana Cavallo, Cristina Costa, Paolo Solidoro

**Affiliations:** 1Respiratory Diseases Unit, Medical Department, AOU Maggiore della Carità di Novara, Corso Mazzini 18, 28100 Novara, Italy; 2Division of Virology, Department of Public Health and Pediatrics, University of Turin, AOU Città della Salute e della Scienza di Torino, 10126 Turin, Italy; 3Division of Virology, Department of Public Health and Pediatrics, AOU Città della Salute e della Scienza di Torino, 10126 Turin, Italy; 4Division of Respiratory Medicine, Cardiovascular and Thoracic Department, AOU Città della Salute e della Scienza di Torino, 10126 Turin, Italy; 5Medical Sciences Department, University of Turin, 10126 Turin, Italy; 6Cardiac Surgery Division, Surgical Sciences Department, AOU Città della Salute e della Scienza di Torino, University of Turin, 10126 Turin, Italy

**Keywords:** lung transplant, cytomegalovirus (CMV), Epstein–Barr virus, herpes simplex virus, varicella zoster virus, human herpesviruses 6 and 7

## Abstract

Lung transplantation is an ultimate treatment option for some end-stage lung diseases; due to the intense immunosuppression needed to reduce the risk of developing acute and chronic allograft failure, infectious complications are highly incident. Viral infections represent nearly 30% of all infectious complications, with herpes viruses playing an important role in the development of acute and chronic diseases. Among them, cytomegalovirus (CMV) is a major cause of morbidity and mortality, being associated with an increased risk of chronic lung allograft failure. Epstein–Barr virus (EBV) is associated with transformation of infected B cells with the development of post-transplantation lymphoproliferative disorders (PTLDs). Similarly, herpes simplex virus (HSV), varicella zoster virus and human herpesviruses 6 and 7 can also be responsible for acute manifestations in lung transplant patients. During these last years, new, highly sensitive and specific diagnostic tests have been developed, and preventive and prophylactic strategies have been studied aiming to reduce and prevent the incidence of these viral infections. In this narrative review, we explore epidemiology, diagnosis and treatment options for more frequent herpes virus infections in lung transplant patients.

## 1. Introduction

Lung transplantation represents a viable treatment option for many end-stage lung diseases [1]; chronic lung allograft dysfunction (CLAD), resulting in bronchiolitis obliterans syndrome (BOS) or restrictive allograft syndrome (RAS), is the most representative cause of mortality after the first year following lung transplantation, thus limiting the long-term survival of transplant patients [2]. On the other hand, infectious complications are the leading cause of death, independently from the time post-transplantation, being responsible for 35% of deaths during the first year and slowly decreasing to 20% thereafter [2]. While bacterial pneumonia and bronchitis are the most common infections, cytomegalovirus and other herpesviruses play an important role in the development of acute and chronic manifestations [3].

Viral infections represent up to 30% of all infectious complications in lung transplant patients [4]. In this context, viruses belonging to the *Herpesviridae*, mainly cytomegalovirus (CMV), Epstein–Barr virus (EBV), herpes simplex virus (HSV), varicella zoster virus (VZV), as well as herpesviruses 6 and 7, play a relevant role; herein, we will review their epidemiologic and clinical features, diagnostic criteria, as well as treatment and prevention strategies. 

## 2. Cytomegalovirus

### 2.1. Epidemiology

Cytomegalovirus (CMV) is a double-stranded DNA virus, with an icosahedral nucleocapsid surrounded by a lipid envelope, belonging to the betaherpesvirus subfamily [5]. CMV is the most common viral pathogen in lung transplant recipients in terms of morbidity and mortality [6], with only bacterial pneumonia being clinically more relevant [7]. After primary infection, the glycoproteins (glycoproteins gB and gH) responsible for virus entry into the target cell trigger the innate immune response, thus engaging the cascade of cytokine production which activates B cells and leads to anti-CMV antibody production [8]. However, due to the variability of viral strains, antibodies are unable to confer adequate protection against donor-derived CMV [9]. In addition, anti-CMV antibodies cross-react with recipient epithelial cells, leading to chronic transplant dysfunction [10].

Primary CMV infection occurs through the direct inoculation with infected cells or body fluids; subsequently, the virus persists in the infected patient for their whole life. Following transplantation, CMV infection may occur as follows [11]: -Transmission from a CMV-seropositive organ;-Transfusion of blood products from a CMV-seropositive blood donor;-Reactivation of a latent infection in a seropositive recipient;-Close physical contact with a CMV-infected individual.

In the general population, CMV seroprevalence is highly variable, reaching 100% in the developing world [12]; the risk of CMV infection among transplant patients is highest in seronegative recipients receiving solid organs from seropositive patients (serologically mismatched donor/recipient, D+/R−) [11]. Other factors that increase the risk of CMV infection are the global immunosuppression level and the type of transplanted organ, with lung transplant recipients being at the highest risk [13].

### 2.2. Clinical Manifestations

A wide spectrum of clinical manifestations of CMV infection has been reported, although not all the infected patients are symptomatic. In fact, CMV infection is defined as the isolation or detection of viral proteins or nucleic acid in any body fluid or tissue specimen independently from the occurrence of symptoms or signs [14,15]; CMV disease is defined as the evidence of CMV infection in the presence of related organ symptoms or signs [11,13]. 

Asymptomatic viremia is the most common presentation in the post-transplant period; it is typically detected during routine evaluation for monitoring the occurrence of CMV infection/reactivation [15].

CMV syndrome is defined as the evidence of clinical disease without end-stage organ involvement [15]; for diagnosing CMV syndrome, the detection of CMV in blood (viral isolation, rapid culture, antigenemia or nucleic acid testing (NAT)) is required with at least two of the following manifestations [15]: -Fever ≥ 38 °C for at least 2 days;-New or increased malaise or new or increased fatigue;-Leukopenia (<3500 cells/µL with total leukocyte count ≥ 4000 cells/µL or a decrease > 20% with total leukocyte count < 4000 cells/µL) or neutropenia (neutrophils < 1500 cells/µL or a decrease > 20% with prior neutrophil count < 1500 cells/µL) on two separate measurements at least 24 h apart;-Greater than or equal to 5% atypical lymphocytes;-Thrombocytopenia defined as platelet count <100,000 cells/µL, a count prior to the development of clinical symptoms ≥ 115,000 cells/µL or a decrease of >20% prior to the development of clinical symptoms to <115,000 cells/µL;-Increase of hepatic aminotransferases (alanine aminotransferase or aspartate aminotransferase) up to twice the upper limit of normal.

CMV pneumonitis is the most common presentation of tissue-invasive diseases in lung transplant patients; clinical manifestations of CMV pneumonitis are non-specific, including fever, dyspnea, dry cough and deterioration of pulmonary function tests; chest radiological alterations include patchy or diffuse ground glass opacities, consolidations, small nodules and combinations of these features (Figure 1) [12,16]. Due to the overlapping of clinical manifestations between CMV pneumonitis and allograft rejection, the distinction of these two alterations may be challenging: as regards the timing of presentation of CMV disease, it rarely occurs within the first two weeks following transplantation, with the highest incidence 55 days after transplantation among patients not receiving CMV prophylaxis [17]; nevertheless, transbronchial lung biopsy (TBLB) is needed to settle the clinical suspicion. Regarding the impact of CMV pneumonitis on patient survival following lung transplantation, it has been demonstrated that patients who did not develop CMV pneumonitis during first 6 months post-transplantation have a better five-year survival rate (71% vs. 53%) compared to those who received antivirals for CMV pneumonitis treatment [18].

CMV has been demonstrated to be associated with CLAD development [18,19,20]; patients who were treated for CMV pneumonitis during the first six months following lung transplantation presented a statistically significantly increased risk of developing CLAD (hazard ratio (HR) 2.19) and post-transplant death (HR 1.89) [18]. As an indirect effect, patients who received CMV prophylaxis had a lower rate of CLAD following lung transplantation [21].

Finally, CMV infection, given its immunomodulatory effects, may predispose to other opportunistic infections, for example, *Aspergillus* species, *Pneumocystis jirovecii*, *Nocardia* and Epstein–Barr virus [12].

### 2.3. Diagnosis

Diagnosis of different CMV manifestations relies on many factors: timing of assay performance, site of sampling and clinical symptoms of the patient. 

-Viral culture: CMV cell culture on a fibroblast monolayer is the only way to evidence virus viability and phenotypically evaluate the occurrence of resistance to antiviral agents; this technique requires skilled and trained personnel. The assay consists in the inoculation of serial dilutions of a BAL fluid specimen on a fibroblast monolayer that is overlaid with a semi-solid medium and the subsequent enumeration of the infected cells. An alternative method for the quantification of CMV in culture is to define the highest dilution of the specimen able to generate a cytopathic effect in 50% of the cultured cells. Nevertheless, these tests are characterized by high specificity but low sensitivity, and they are laborious and time consuming [22]. Shell vial assay reduces the time for viral detection, but it is not widely used. This assay requires a low-speed centrifugation of BAL fluid on a fibroblast monolayer combined with the detection of viral antigens produced at the early stages of infection by monoclonal antibodies, paired with the use of a second antibody labeled with a fluorescent dye or an enzyme, with viral load being estimated by the number of labeled cells in a microscopic count. Shell vial assay allows for obtaining results within 24–48 h, thus shortening the time needed to detect a positive signal and limiting the contact of cells with toxic products usually present in clinical specimens [13,22]. Compared to traditional viral isolation, shell vial assay based on indirect immunofluorescence (IF) targeting CMV immediate early antigen (p72) reduces the time for viral detection. However, techniques based on viral isolation are now largely replaced by molecular assays (Figure 2) [23]-*Antigen assay:* It consists in the semi-quantitative detection of pp65-infected polymorphonuclear cells by direct immunofluorescence staining. It has been demonstrated that both pp65 antigenemia and CMV-DNA detection in blood or plasma correlated with CMV infection and disease, although discrepant results may occur [12]. Moreover, antigenemia assay requires a high level of training and expertise and is highly resource consuming [24]. It is now considered an obsolete test and its efficacy is limited in patients with leukopenia [25].-*Serology:* Serology testing (CMV-specific IgG) is fundamental in the pre-transplant setting for D/R CMV risk stratification [13]; in recipients seronegative before transplantation, seroconversion may occur after receiving blood transfusions or immunoglobulins; repetition of IgG testing is suggested within two months before lung transplantation [26]. The use of CMV-specific IgM is very limited because of the immunosuppressive effect following lung transplantation [27].-*Nucleic acid testing:* CMV RNA or DNA is preferred for detection of viral replication [13]; both whole blood and plasma can be used, with viral load usually being one log lower in plasma in comparison to whole blood [28]. Therefore, laboratories that evaluate viral load in plasma use highly sensitive tests to detect low levels of CMV viremia [29]. Quantitative nucleic acid testing (QNAT) may evidence interlaboratory variability and lack of standardization; to overcome this, however, in 2010 the World Health Organization standardized CMV QNAT [30]. This facilitated comparison and allowed the establishment of a threshold for the management of antiviral therapy [31]. More recently, consensus guidelines suggest the use of doubling time of viral load as the most reliable tool for deciding the administration of pre-emptive anti-CMV therapy [32].

Diagnosis of different CMV manifestations is established by the 2018 International.

Consensus Guidelines on CMV Management in Solid Organ Transplantation [33]:-Asymptomatic viremia: When CMV viral load exceeds the threshold used as the limit for administration of antiviral drugs [33]; this clinical manifestation is usually found during routine post-transplantation monitoring;-CMV syndrome: When CMV viral load exceeds the threshold in patients with a clinically compatible syndrome (malaise, fever, leukopenia and/or lymphocytosis in the absence of end-organ disease) [33];-CMV pneumonitis: Detection of CMV by QNAT in bronchoalveolar lavage (BAL) fluid reflects CMV replication in the lung and the increase in DNA levels correlates with symptomatic CMV disease [33]; qualitative detection of CMV DNA in BAL is not specific for CMV pneumonitis diagnosis [34]. When clinically feasible, TBLB can distinguish CMV pneumonitis from acute rejection; a definitive diagnosis is made when CMV inclusion bodies are present in biopsies and/or CMV viral antigens are found during immunohistochemistry or DNA hybridization tests are positive on lung tissue biopsies [33]. In the case of the impossibility to obtain allograft lung tissue, a presumptive diagnosis of CMV pneumonitis can be based, in an appropriate clinical and radiological setting, on the detection of virus with QNAT or CMV culture in BAL fluid [33], nevertheless, it should be noted that this approach cannot differentiate viral shedding from tissue-invasive disease.

Recently, a consensus statement for the standardization of BAL in lung transplantation has been published by the International Society for Heart and Lung Transplantation (ISHLT) [35]; this document arises from the need to uniform the variability in interpretation of BAL results, bronchial wash (BW) and related definitions and indications. BAL can be performed as a surveillance (scheduled follow-up protocol) or diagnostic test (based on clinical indication). Surveillance BAL usually identifies asymptomatic infection in 12% to 40% of cases, particularly 6 to 12 months post-transplantation [35]. CMV PCR is widely performed on BAL fluid, however, only some centers routinely perform this evaluation (nearly 60%) [35].

### 2.4. Risk Factors

CMV serological status is the most important risk factor for CMV infection and disease in lung transplant patients; moreover, the use of immunosuppressive therapies plays an important role in CMV replication. 

-CMV serology: as already mentioned, CMV donor (D) and recipient (R) serostatus is evaluated during pre-transplantation evaluation to define the risk of developing CMV infection.○D+/R−: this group of recipients presents the highest risk for CMV manifestations; CMV pneumonitis occurs in up to 85% of recipients even if treated for 2–3 weeks with ganciclovir and standard intravenous immunoglobulins [36]. The introduction of more effective antiviral agents has reduced the historically reported fatality rate of 22% in this subgroup of recipients [37].○D−/R+ and D+/R+: the incidence of CMV infection is lower for these subgroups of patients, reaching 58% and 69%, respectively [38]. In D+/R+, the risk is higher due to the reactivation of latent virus and superinfection with a new infecting strain. More recent studies, performed in the era of new antiviral agents, demonstrated that the incidence of CMV infection or disease in these subgroups of patients is much lower [39].○D−/R−: the risk of developing CMV infection is the lowest in seronegative recipients with seronegative donors.-CMV-specific immunity: more recently, several assays have been developed to evaluate CMV-specific T cell response; these tests are based on the evaluation of activation of T cells in response to specific antigenic stimulus. T cell reactivity is quantitatively evaluated by measuring ex vivo production of IFN-γ following incubation with CMV-specific peptides [12]. These methods are commonly indicated as IFN-γ-releasing assays (IGRA) and include: ○QuantiFERON^®^-CMV assay (Cellestis Ltd., Melbourne, VIC, Australia): this test utilizes an IFN-γ semi-quantitative technique to assess CMV immune reconstitution [12]. A prospective multicenter cohort study demonstrated that D+/R− QuantiFERON^®^-CMV-positive patients, who received antiviral prophylaxis for a median of 98 days, at 12 months post-transplantation had a significantly lower risk of CMV disease (6% versus 22% among QuantiFERON^®^-CMV-negative patients) [40]. A very recent retrospective analysis on lung-transplanted patients evaluated different outcomes based on QuantiFERON^®^-CMV direct prophylaxis (5 or 11 months) compared to standard of care (5 months): researchers found that patients who received QuantiFERON^®^-CMV direct prophylaxis had lower incidence of CMV infections, in particular those who received an extended prophylaxis [41].○CMV-specific linked immunospot (ELISPOT): this assay is based on the measurement of IFN-γ produced by T cells in response to immediate early and/or pp65 antigens (Figure 3) [12]. ELISPOT, as well as QuantiFERON^®^-CMV, predicts CMV viremia [42]; in lung transplant patients, CMV disease with a longer duration of infection occurs earlier in non-responders (recipients with a negative test) than responder patients [43]. Moreover, ELISPOT response appears to predict a lower incidence of CMV viremia, with a change in this status following lung transplantation in patients who received a combined CMV prophylaxis with antiviral agents and CMV-specific immunoglobulins [44].

### 2.5. Prevention of the Disease

Due to the relevant morbidity and mortality, as well as the increased risk of CLAD related to CMV infection, two different preventive strategies have been used: -Universal prophylaxis: it is the most widely used strategy for CMV disease prevention, with administration of antiviral agents (oral valganciclovir and intravenous ganciclovir) alone or in combination with CMV immunoglobulins [33,45]. The approach to prophylaxis strategy may differ among transplantation centers, however, it is based on international guidelines [33]. This prophylaxis scheme provides the intravenous administration of ganciclovir (5 mg/kg once daily, with adjustments of the dose based on renal function) in all recipients who are seropositive for CMV or who received an organ from a seropositive donor (D+/R+, D−/R+, D+/R−). Ganciclovir can be switched to valganciclovir 900 mg once daily when the recipient can receive drugs orally. Most recent international consensus guidelines recommend a prophylaxis for 6 to 12 months for CMV D+/R− recipients and a minimum of 6 months for D+/R+ and D−/R+ patients [33]. Different durations are based on CMV reactivation risk, drug toxicity development and viral load monitoring schemes [33]. Valganciclovir has been demonstrated to be effective in CMV infection and disease prevention in lung-transplanted patients; a randomized trial demonstrated that valganciclovir, administered for 12 months, reduced the incidence of CMV disease and infection in comparison to patients who received valganciclovir for 3 months (4% vs. 32% and 10% vs. 64%, respectively), with no increased risk of ganciclovir-resistant CMV infection [46]. Other authors tested prophylaxis with ganciclovir for 6 months and found that development of CMV infection or disease differed depending on serostatus, with higher incidence among CMV D+/R− recipients [38]. Ganciclovir has been replaced by valganciclovir as drug of choice for CMV prophylaxis; ganciclovir inhibits viral replication, although it does not eradicate the latent infection. For this reason, ganciclovir prophylaxis is effective, but it does not provide long-term protection [47]. A survey conducted in 2010 among lung transplant centers evidenced that most centers used the universal prophylaxis scheme although with variable duration, ranging from 3 to 6 months based on CMV D/R serostatus [48]. Neutropenia is the most common and significant side effect associated to valganciclovir and ganciclovir use; for this reason, the monitoring of blood cell count, as well as renal function, is suggested in order to guide appropriate antiviral dosing. CMV immunoglobulins represent an additional therapy to antiviral agents, although this approach is still controversial. Some centers use CMV Ig in addiction to antiviral therapy in high-risk patients (CMV D+/R−, D+/R+, D−/R+ and in recipients treated with a lymphocyte-depleting agent), evidencing an increased duration of the period free from CMV infection and disease, but the effects directly related to CMV Ig are not demonstrable [49]. Another retrospective study demonstrated an increased efficacy in preventing CMV infection sequelae by CMV Ig added to ganciclovir in comparison to ganciclovir alone [50]. Specific circumstances supporting the use of CMV Ig occur in patients with prolonged neutropenia who are intolerant to ganciclovir, those with refractory CMV disease and hypogammaglobulinemia [32].-Pre-emptive therapy: with this strategy, patients are monitored with a weekly (or longer) CMV load measurement; antiviral therapy is administered only when viral replication exceeds a threshold defined in each institution [32]. This strategy detects early viral replication, thus preventing progression to clinical disease [32]. As specified above, due to the high intertest variability in different centers, a universal threshold for the initiation of antiviral administration cannot be defined. Due to its nature, this strategy allows for the administration of antiviral drugs to a lower percentage of patients and the course of treatment is shorter; this decreases the drug use and exposure to side effects, while reducing costs and the occurrence of drug resistance [32]. When indicated, the treatment should be based on valganciclovir 900 mg orally twice a day or intravenous ganciclovir 5 mg/kg every 12 h for a minimum of 2 weeks and until the viral load is undetectable (or below quantifiable limits) [32].

Several trials compared universal prophylaxis to a pre-emptive strategy in renal recipients and reported no difference in the incidence of CMV disease or intragraft CMV infection [32]. To date, no trial that has directly compared the two different approaches in lung-transplanted patients has been published; potential benefits of each scheme are reported above. 

### 2.6. Treatment of the Symptomatic Disease

The treatment of symptomatic cases consists in a combination of antiviral drug administration and reduction of immunosuppression. 

-Antiviral agents: both valganciclovir and ganciclovir can be used for the treatment of symptomatic CMV disease. ○CMV syndrome: the majority of CMV syndrome cases have a mild to moderate infection that can be managed with oral valganciclovir, with dose of 900 mg twice daily. In case of impossibility to take oral medication or rapidly progressive or severe disease, intravenous ganciclovir is preferred at 5 mg/kg twice daily [32]. Viral load should be monitored weekly or every 2 weeks; the treatment is administered until the viral load is undetectable or it is below the quantifiable limit of the assay (on two consecutive detections in the case of less sensitive assays) [32]. The duration of therapy should not be less than 2 weeks [51]. In the VITOR trial (lung-transplanted patients represented 6% of the study population), valganciclovir was compared to intravenous ganciclovir for the treatment of CMV disease in solid organ transplant recipients; the treatment in each group was continued for 21 days and then switched to valganciclovir 900 mg once daily for another 28 days. No difference was reported in the clearance of CMV viremia at day 21 of therapy (45% vs. 48% respectively), neither for clinical resolution of CMV disease (77% vs. 80% respectively) nor for other long-term outcomes (clinical or viral eradication of CMV disease, incidence of ganciclovir resistance at one year) [52]. ○CMV pneumonitis and tissue-invasive disease: intravenous ganciclovir is preferred for the initial treatment of severe and tissue-invasive disease, although bioavailability of valganciclovir is similar [32]. As for CMV syndrome, intravenous ganciclovir at 5 mg/kg twice daily with dose adjustments based on renal function is recommended, shifting to valganciclovir with improvement of clinical conditions and possibility to take oral drugs [32]. The duration of therapy is based on the clinical response and CMV viral load reduction (monitored weekly or every 2 weeks). The treatment should be continued until the viral load is undetectable or it is below the quantifiable threshold of the assay (two consecutive measurements in the case of less sensitive assays) [32].-Reduction of immunosuppression: the dose of immunosuppressant should be tapered to support the generation of CMV-specific immunity [53]. This decision should be considered in patients with more severe disease, high viral loads, slow clinical and virological response and ganciclovir-resistant CMV [32,53]

## 3. Epstein–Barr Virus

### 3.1. Epidemiology and Clinical Manifestations

Epstein–Barr virus (EBV) is a ubiquitous Ɣ-herpesvirus that, after primary infection, is able to persist in the host for life [54]. More than 90% of adults and 50% of children globally are seropositive for EBV [54,55]. Primary EBV infection, transmitted by saliva, is usually acquired in childhood. Most EBV infections in children are asymptomatic while, in approximately 50% of adolescents and adults, a syndrome of infectious mononucleosis may develop [56]. Rarely, EBV-infected patients may develop fatal infectious mononucleosis [57,58,59], chronic active infection [60,61,62], hemophagocytic syndrome [57] or neoplasms such as Burkitt’s lymphoma, undifferentiated nasopharyngeal carcinoma, gastric carcinoma, Hodgkin’s lymphoma, T cell lymphoma, nasal NK/T cell lymphoma, aggressive NK/T cell lymphoma/leukemia, leiomyosarcoma and lymphoproliferative disorders in immunocompromised hosts, including AIDS-associated lymphomas [62,63,64,65].

The great importance of EBV among transplant recipients originates from its ability to transform infected B cells, thus determining post-transplantation lymphoproliferative disorders (PTLDs), a life-threatening complication. Lymphoproliferative disorders account for about 20% of all post-transplant malignancies, in comparison with only 5% in the general population [66]. In hematopoietic stem cell transplant (HSCT) recipients and in solid organ transplantation (SOT), EBV infects naïve B cells to transform them into proliferating blasts, potentially resulting in EBV PTLDs. In particular, in lung transplant recipients, PTLD typically presents within the first year post-transplantation with a predilection for the lung allograft in 70–90% of cases and for the abdomen in 20–35% [56,67,68,69,70]. Epstein–Barr-virus-infected B cells in SOT recipients are usually of recipient origin and account for more than 90% of PTLDs complicating SOT, whereas PTLDs in patients undergoing allogeneic hematopoietic cell transplantation (HCT) arise primarily from the donor [71,72]. Unlike HSCT, only about half of PTLDs in SOT are EBV positive, and the majority of late-presenting PTLDs are EBV negative [73].

Management of EBV PTLD is more complex, involving pre-emptive measures, EBV DNAemia evaluation and a correct balancing between treatment options such as, for example, reduction of immune suppression, anti-B-cell therapy and, more recently, cytotoxic T lymphocytes (CTLs) [73]. In transplant recipients, immunosuppressive drugs suppress the function of EBV-specific CTLs, allowing uncontrolled proliferation of EBV-infected B cells. Development of PTLD is associated with elevated EBV DNA loads and a deficiency of EBV-specific CTLs [74].

### 3.2. Risk Factors 

Donor/recipient EBV serostatus, intensity of immunosuppression and organ transplant are the most important risk factors for PTLD [66,72,73,75,76]. The incidence of PTLD is highest in EBV-seronegative recipients who are at risk for primary EBV infection following transplant in comparison to seropositive recipients (relative risk 2.6 versus 9.9, respectively) [72,77,78,79,80]. Transplantation of an EBV-positive organ in a seronegative recipient is associated with a severalfold increase in risk of PTLD. Pediatric populations have a higher incidence of PTLD due to a higher proportion of EBV-naïve individuals [79,81]. Moreover, the incidence of PTLD varies by organ transplanted, reflecting differences in the degree of immunosuppression. Incidence rates of PTLD are highest among recipients of intestine (up to 20%) allografts, followed by lung (3–10%), heart (2–8%), liver (1–5%) and kidney (0.8–2.5%) [82]. Lympholytic antibodies such as antilymphocyte globulin (ALG) and antithymocyte globulin (ATG), T cell depletion and high EBV viremia represent other risk factors [75,83,84]. Moreover, coinfection with CMV may play a role in the development of PTLD [75,85].

### 3.3. Diagnosis

Early detection of EBV viremia and optimal interventions with either pre-emptive therapy and/or a reduction of immunosuppression may significantly impact on transplant outcomes. Surveillance of EBV DNAemia using molecular assays remains the standard approach in preventing PTLD in high-risk populations [73,74,86]. In particular, multiplex PCR assays consisting of EB-encoded RNA 1 (EBER-1), latent membrane protein 1 (LMP-1) and EBNA-2 are the most sensitive for diagnosis and monitoring response to treatment for EBV-related PTLD [56,67,71,87]. However, which of the blood compartments, including plasma, whole blood or peripheral blood mononuclear cells (PBMCs), is optimal for testing is unclear [88]. There is also a large variation in practice regarding which optimal EBV load cut-off to use for initiating and interrupting pre-emptive therapy [89]. Lazzarotto and colleagues, investigating the kinetics of CMV and EBV DNA in whole blood and plasma of SOT/HSCT recipients, observed that whole blood seems the most reliable specimen type for the post-transplant surveillance of both active CMV and EBV infections [90,91]. Moreover, Kanakry and colleagues observed that the detection of EBV DNA in plasma was a better marker of EBV+ PTLD than PBMCs [92]. These findings highlight the clinical importance of frequent EBV DNA load monitoring in lung transplantation recipients. In particular, SOT recipients who are EBV seronegative prior to transplant should be frequently monitored for EBV DNAemia at regular intervals following transplant. Reduction of immunosuppression has been shown to reduce the incidence of early PTLD in pediatric SOT recipients [93]. Likewise, the reduction of immunosuppression during primary EBV infection promotes the development of EBV-specific T cell responses and it could be used as a pre-emptive strategy in adult and pediatric EBV-seronegative SOT recipients [73].

### 3.4. Therapy and Prevention

Antiviral agents that target lytic EBV infection (e.g., acyclovir, valacyclovir, ganciclovir, foscarnet) are used to prevent and to treat PTLD [94]. However, antiviral prophylaxis and pre-emptive therapy, particularly with acyclovir and ganciclovir, have not been effective in preventing PTLD in SOT or HCT recipients [73,74]. In fact, these antiviral agents are efficacious during the acute phase of EBV infection, while latent EBV in memory B cells, characteristic of PTLD, is not affected by lytic agents. Nevertheless, antiviral therapy, coupled with minimization of immunosuppression, is sometimes adequate for early PTLD. Regarding new prevention strategies, “pre-emptive” therapy with rituximab or adoptive cellular immunotherapy using EBV-specific cytotoxic T lymphocytes may reduce the risk of PTLD in asymptomatic HSCT recipients with a high EBV load in the blood [95]. However, this approach is labor intensive and expensive, available only in selected research centers.

## 4. Herpesviruses 1 and 2

### 4.1. Epidemiology and Clinical Manifestations

Herpes simplex virus (HSV) is a double-stranded DNA virus belonging to the α-herpesvirus family and encompasses two subtypes (HSV-1 and -2) with seroprevalence in occidental countries ranging from 50% to 88% and 3% to 23%, respectively [96,97], although rates have declined over the past two decades. Transmission occurs by direct contact with infected skin or secretions. In immunocompetent adults, HSV-1 is typically associated to oral mucocutaneous herpes, whereas HSV-2 causes anogenital disease; however, epidemiological changes have been increasingly evidenced with genital and neonatal herpes caused by HSV-1 and orolabial lesions by HSV-2 [97,98]. Following primary infection, HSV establishes latent infection of sensory ganglion neurons, from which virus may reactivate with asymptomatic mucosal shedding (e.g., HSV-1 has been isolated from the saliva of 1–5% of healthy subjects [99]) and recurrent vesicular disease, often through autoinoculation, with ocular and skin involvement. Severe diseases are uncommon in immunocompetent subjects, with rare description of hepatitis and neurological syndromes, including meningitis, Bell palsy and encephalitis, with significant morbidity and mortality [100,101]. 

In the critical and/or immunocompromised host, HSV has been associated with severe diseases, including visceral or disseminated disease with extensive mucocutaneous involvement, hepatitis, meningoencephalitis and pneumonitis potentially associated with poor outcome and high mortality rates, particularly in the presence of high viral load [99,102,103,104].

In transplanted recipients, HSV reactivation occurs frequently with overall incidence of infections being 18–30% [103], typically during the first months post-transplantation [105] because of reactivation from latency within the transplant recipient, though transmission via infected donor organ has been reported [106,107,108]. 

In transplant patients, most typical manifestations of HSV disease consist in mucocutaneous syndromes, often with prolonged duration of symptomatology and local extension (i.e., involvement of lower respiratory tract or esophagus). Disseminated infections, in particular hepatitis, are uncommon although burdened with significant mortality and can occur with tracheobronchitis and concomitant pneumonia displaying extensive necrosis with diffuse infiltrates on radiographic imaging and infected cells with intranuclear inclusions [109,110,111]. 

Data regarding HSV infections, specifically among lung transplant recipients, are scant in the literature. Although severe infections are relatively uncommon, even asymptomatic viral shedding may be associated with increased mortality [106]. Nagarakanti and colleagues reported a case of adenovirus, HSV and CMV infection in a lung transplant recipient with fatal outcome despite some initial amelioration with antiviral administration [112]. A case of HSV-2 hepatitis has been reported four years after lung transplantation in a patient with cystic fibrosis; despite the delay in diagnosis, the patient fully recovered with acyclovir, reduction of immunosuppression and intravenous immunoglobulin [113], thus underlining that rapid treatment with high-dose acyclovir is critical as the disease may be fatal if untreated. 

### 4.2. Diagnosis

Diagnosis can be made by classic methods such as viral culture [114] or antigen testing, although molecular assays for detection of HSV DNA are preferred due to the increased sensitivity and rapidity, therefore representing the gold standard. Depending on the type of specimens, quantitative assays can be useful for evaluating and monitoring infection/disease, response to antiviral therapy and outcome. Specimens are usually collected in relation to the clinical context, including secretions from vesicular lesions, swabs or biopsies, bronchoalveolar lavage, cerebrospinal fluid or whole blood. A potential role of multiplex molecular assays for viral and non-viral pathogens should be considered because of enhanced pathogenicity by either virus–virus interaction or virus–host interaction due to modulation of the host cell immunity or production of cytokines [115].

Moreover, considering the role played by specific cellular immunity in controlling latency and viral replication, assays evaluating IFN-gamma release (IGRA) from virus-specifically activated T cells could be useful in the tailored clinical–therapeutic management of transplant patients, as already reported for some herpesviruses, including CMV, EBV and HSV [116,117,118,119,120]. Quantitative evaluation of HSV-1-specific T cell response in the blood compartment, and the study of the relation between this and the ability to control local reactivation in the lung, could be relevant in transplant patients at risk of severe pulmonary complications. By using a home-made developed IGRA assay for HSV, a lower response in the first three months post-transplantation with a progressive recovery of pre-transplantation status by the second year and in the presence of HSV-1 DNA positivity in bronchoalveolar lavage was shown. 

### 4.3. Therapy and Prevention

As regards treatment of HSV infections/diseases, as for other transplant settings, intravenous administration of acyclovir is the preferred treatment in case of severe infections, whereas oral acyclovir, valacyclovir (prodrug of acyclovir) or famciclovir (prodrug of penciclovir) can be used in the presence of limited manifestations. Acyclovir resistance (also implying resistance to famciclovir/penciclovir), although uncommon, can occur in immunocompromised patients potentially having received multiple courses of antivirals due to multiple recurrences. In these cases, foscarnet or cidofovir is used as an alternative. Prophylaxis is often administered in transplant patients, using acyclovir or valacyclovir; moreover, it is to be taken into consideration that prophylaxis for CMV with valganciclovir is also effective against HSV.

## 5. Varicella Zoster Virus (VZV)

### 5.1. Epidemiology and Clinical Manifestations

The varicella zoster virus (VZV) belongs to the Herpesviridae family [121], is a neurotropic human α-herpes virus and is one of the most common causes of opportunistic infections among hematopoietic stem cell transplant (HSCT) [122,123,124] and solid organ transplant (SOT) recipients [125,126,127,128]. A particular case of fatal septic shock after VZV reinfection in a lung-transplanted patient was described by Lehingue et al. [129]. The patient (62 years old) was VZV seropositive and was admitted to an intensive care unit after lung transplantation complicated by graft dysfunction. After a few days, a screening for VZV infection by quantitative blood polymerase chain reaction was positive with a viral load of 4.2 × 10^4^ copies/mL. Despite of the administration of pharmacological therapies, the patient developed multiorgan failure, never showed skin eruption and died after nine days. The donor was seronegative for VZV; however, VZV-specific molecular tests on donor lung biopsies and blood samples were performed and were positive. It is likely that the fatal evolution was correlated with the disseminated VZV infection without skin eruption in the context of lung transplantation, as reported by Carby et al. [127]. Primary varicella infection presents with febrile illness and a widespread skin rash with a self-limiting course in immunocompetent children [121,130]. After resolution of the initial infection, VZV establishes latency within the sensory ganglia and, subsequently, in individuals with compromised immunity, the virus can reactivate, giving rise to stereotyped vesicular skin lesions, termed herpes zoster (HZ) [130,131]. HZ is characterized by a painful vesicular rash limited to a single dermatome. Diffuse cutaneous dissemination, severe pneumonia, encephalitis and visceral involvement are rare complications and may occur in immunocompromised patients [132].

More than 90% of adults in the United States are VZV positive, and there are over one million cases of HZ each year [124,130,132]. HZ is rare in immunocompetent individuals, but the incidence increases with age, rising to 10 cases per 1000 patient-years by age 75 [131]. The incidence and severity of VZV infections increase in immunocompromised patients; indeed, in one study of transplant patients, the incidence of HZ was 175 cases/1000 person-years (cumulative incidence rates of 27% at 1 year; 36% at 2 years; 44% at 3 years) [124,133].

Most cases of VZV present vesicular skin lesions accompanied by itching, tingling, dysesthesia or pain. The rash is erythematous and progresses to pustules, ulcers and crusting which subsequently disappear [130]. Dissemination occurs in up to 10% of transplant patients, and visceral involvement and death are rare (<2%) [122,128,134]. Some complications from VZV can be meningitis and meningoencephalitis, ocular manifestations such as keratopathy, iritis or acute retinal necrosis [135]. Secondary bacterial infections can develop at the site of vesicular skin lesions. Long-term complications can be chronic and debilitating (motor impairment, post-herpetic neuralgia, ophthalmic HZ) [136,137]. Post-herpetic neuralgia occurs in 20–25% of patients with HZ; recurrent HZ infection is rare (<2%), but recurrence rates in transplant recipients or children with ALL are 5–9.7% [137].

### 5.2. Diagnosis

The diagnosis of VZV is usually established based on signs and symptoms. Direct immunofluorescent (IF) assays and viral cultures are less sensitive than molecular tests [114]. In atypical forms or in monitoring immunocompromised patients, the virus can be detected by PCR in blood or tissue [123,130].

### 5.3. Risk Factors

The risk of VZV infections reflects the serological status and the intensity of immunosuppression (in transplant patients). Risk factors for VZV and HZ include immunosuppression with high-dose corticosteroids [138], HIV infection [139], acute lymphoblastic leukemia [122], multiple myeloma [140], chemotherapy for hematologic malignancies [138], HSCT [138] or SOT [125,126,127,141]. Among SOT recipients, incidence rates include 3 to 10% (kidney), 5.7 to 12% (liver), 6 to 16.8% (heart) and 12. 5 to 20.2% (lung) [141,142,143,144,145].

### 5.4. Therapy and Prevention

Acyclovir (oral or IV), valacyclovir (oral) and famciclovir are the most commonly used therapies for complicated VZV infections [146]. Ganciclovir and valganciclovir demonstrated good in vitro activity against VZV. Foscarnet may be effective as therapy for acyclovir-resistant VZV, but resistance to foscarnet can develop [147,148].

Prophylaxis with acyclovir or valaciclovir is effective in high-risk immunocompromised patients [149,150]. A live attenuated VZV vaccine (Oka virus, Zostavax, Merck, Readington Township, NJ, USA) has been shown to reduce the frequency and severity of VZV infections and post-herpetic neuralgia in the elderly and at-risk populations [151,152,153]. This vaccine is distinct from the standard varicella vaccine (Varivax, Merck), has 14-fold higher antibody titers and is designed to stimulate waning immunity in the elderly, however, live viruses are contraindicated in immunosuppressed individuals, while in pre-transplant candidates seronegative for VZV the standard varicella vaccine (Varivax, Merck) is suggested [154,155,156]. Recombinant zoster vaccine (RZV, Shingrix, GSK) was licensed in the United States for the prevention of herpes zoster in adults aged ≥18 years who are or will be at increased risk for shingles because of immunodeficiency or immunosuppression caused by known disease or therapy [157]; a study conducted on kidney-transplanted patients demonstrated that recombinant zoster vaccine reduced incidence of herpes zoster manifestations (incidence in the vaccinated and unvaccinated groups was 3.9% and 13.7%, respectively) [158].

## 6. Human Herpesvirus 6 and Human Herpesvirus 7

### 6.1. Epidemiology and Clinical Manifestations

Human herpesviruses 6 (HHV-6) and 7 (HHV-7) belong to the Herpesviridae family, genus Roseolovirus [159]. They establish lifelong latent infections, and their genomes are closely related to each other and to that of CMV [160]. In immunocompetent patients, infections are usually self-limiting; among immunocompromised hosts, such as HSCT and solid organ transplant (SOT) recipients, they are responsible for serious direct and indirect effects.

HHV-6 was first isolated in 1986 in the USA from patients with lymphoproliferative disorders [161]. HHV-6 has particular tropism for CD4 lymphocytes [160,162], and, in vivo, it was isolated from lymph nodes, endothelial cells, monocytes, macrophages, tonsils, salivary glands, kidney, liver, lung and the central nervous system (CNS) [163]. Monocytes and macrophages are latency sites [164]. HHV-6 comprehends two distinct species, HHV-6A and HHV-6B [160,162]. It is the only human herpesvirus able to integrate into the host chromosome and to be vertically transmitted [164].

HHV-7 is a lymphotropic virus [165], first discovered in 1990 in the USA [166] in T cells of a healthy adult. It was isolated from salivary glands, skin, lymphoid tissue, liver, kidney and lungs [163]. It establishes latency in CD4 lymphocytes [163]. HHV-7 integration into the human genome has been hypothesized [163]. The presence of telomeric repeats sequences (TRSs) in HHV-7 as in other herpesvirus genomes suggests the possibility of homologous recombination between the telomeric region of human chromosomes and viral TRSs, facilitating the integration. However, at this moment, only HHV-6 integration into human chromosomes has been proven [163].

They are ubiquitous, and their seroprevalence is >90% in adults [167]. Saliva is the main transmission route at a young age, whereas, in adults, organ transplantation plays an important role [163,168,169]. HHV-6 primary infection is frequent in the first 2 years of life with a peak of incidence from 6–9 months of age [170]. The prevalence of chromosomally integrated HHV-6 (ciHHV-6) is estimated to be 1% in healthy individuals [160]. The majority of infections are caused by HHV-6B, while HHV-6A epidemiology is less well known but is considered predominant in Africa [162]. In transplant recipients, HHV-6 reactivation was observed in 45% of HSCTs and 30% of SOTs [171], particularly 2–4 weeks post-transplantation, with a predominance of HHV-6B [168,172].

HHV-7 primary infection is generally acquired in childhood within 5 years of age [173]. HHV-7 reactivation usually occurs after 1 to 4 weeks in 40% of HSCT and SOT recipients, with brief and moderate clinical manifestations [163,171,174,175].

In normoergic young patients, HHV-6 is typically associated with exanthema subitum. Other clinical manifestations are fever, seizures, rash and gastrointestinal and respiratory signs [162,172,176]. Adult infections are uncommon, although HHV-6 reactivation could be triggered by immunodepression. Clinical manifestations are almost exclusively due to HHV-6B [163]. In SOT, HHV-6 is associated with increased mortality and the reactivation rate varies according to organ transplant, immunosuppressive regimen and anti-CMV prophylaxis, which is supposed to be protective against HHV-6 [163,177]. HHV-6A’s prevalence is limited to 3% [171], whereas ciHHV-6’s prevalence is still poorly known [163]. In lung and heart transplants, HHV-6 is associated with direct effects such as fever, rash, myelosuppression, encephalitis, hepatitis and pneumonitis [163,175,178,179,180,181]. HHV-6 reactivation is also associated with indirect effects: allograft rejection, CMV replication and fungal and other opportunistic infections [175,181].

HHV-7’s clinical manifestations are not well characterized; they are similar to HHV-6B’s but less frequent [163]. In transplant recipients, HHV-7 reactivation is associated with myelosuppression, CNS disease, hepatitis, pneumonitis and increased risk of CMV disease. However, HHV-7’s role in direct and indirect effects is difficult to discern, especially in the presence of CMV reactivation, due to the close interaction between both viruses [182].

### 6.2. Diagnosis

Serology is restricted to primary infections and seroprevalence studies [163]. Viral culture and antigen detection have limited application in routine diagnostics [163]. The SOT diagnostic reference method for active infection is based on nucleic acid amplification tests of clinical samples [163,177]. Viral monitoring in whole blood, BAL fluid (BALF) and lung biopsies is useful in lung transplant patients’ management even if no thresholds have been established [163,177,178]. Guidance about surveillance tests is still lacking and viremia monitoring is not recommended in SOT recipients’ management, nor in prophylaxis or in pre-emptive therapy [163,175,177].

### 6.3. Therapy

Ganciclovir, valganciclovir, foscarnet and cidofovir have in vitro activity against HHV-6 and HHV-7, but in vivo studies are lacking [163,177]. Treatment options are limited to selected cases such as immunocompromised patients with severe clinical presentation [174,183].

## Figures and Tables

**Figure 1 viruses-15-02326-f001:**
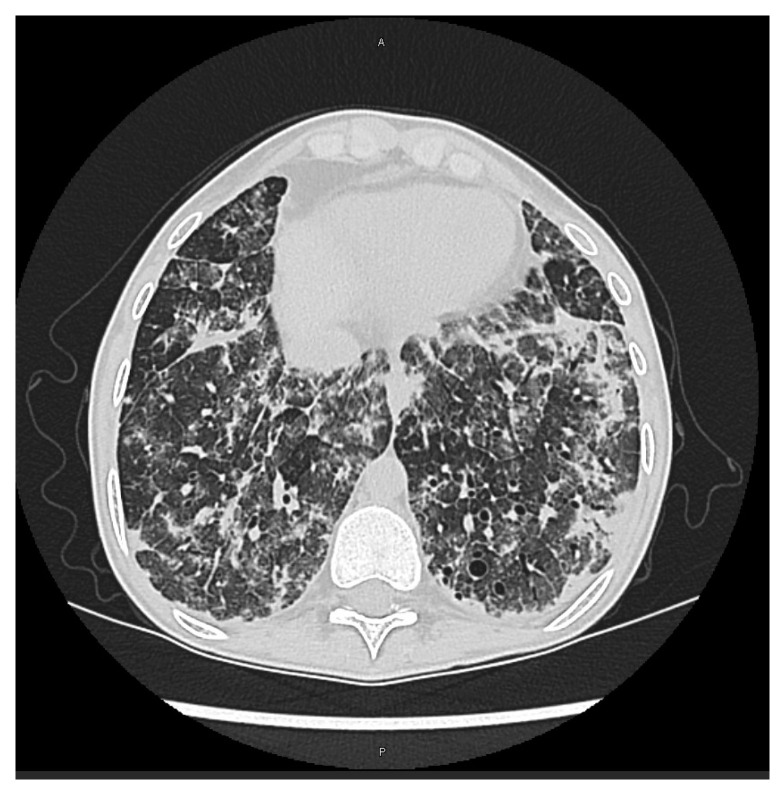
CT scan of cytomegalovirus bilateral pneumonitis in a lung-transplanted patient for cystic fibrosis: bilateral consolidations and ground glass opacities in lower and middle lobes and lingula.

**Figure 2 viruses-15-02326-f002:**
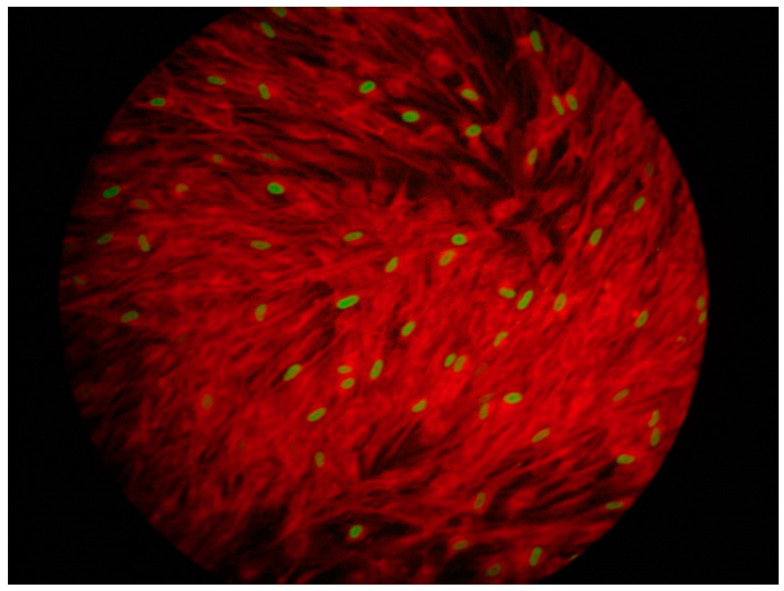
Rapid shell-vial viral isolation for CMV on cell culture of human embryony lung fibroblasts (after 24 h of incubation): indirect immunofluorescence (IF) test. Nuclei of cells (human embryony lung fibroblasts) infected by CMV appear green/yellow, while the non-infected nuclei and cytoplasm appear red.

**Figure 3 viruses-15-02326-f003:**
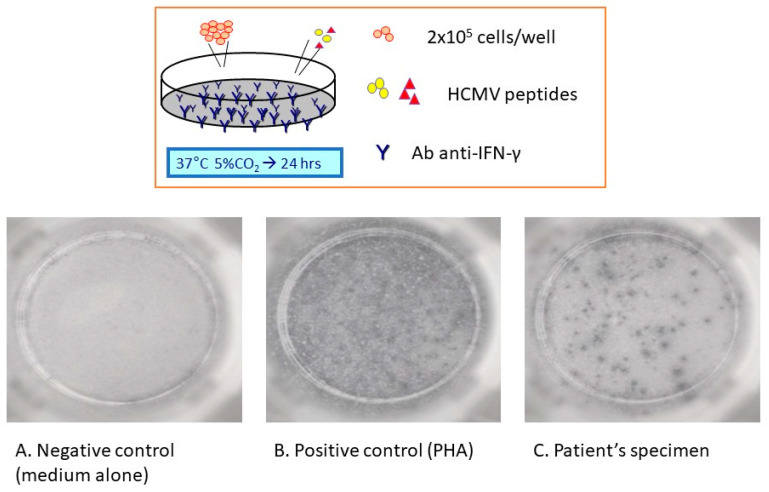
ELISPOT assay: 200,000 cells were incubated with HCMV peptides in microplate wells coated with anti-IFN-gamma antibodies at 37 degrees, 5% CO_2_ for approximately 24 h. Then, an immunoenzymatic assay was used to evidence the IFN-gamma-secreting cells as single spots. Microplate reading of the number of IFN-gamma-secreting cells was carried out by a computer-assisted image analysis system. Results are both qualitative in terms of responder (recipients with a positive test) versus non-responders (recipients with a negative test) and quantitative in terms of the number of spot-forming units per well. According to previously published studies, no response was defined as a number of SFU/well lower than 5, a weak response between 5 and 20, a good response higher than 20. Temporal profile was considered as early when reconstitution of immune response occurred within 30 days post-tx and late after 30 days.

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
