# Peer review of "Herpes Virus Infection in Lung Transplantation: Diagnosis, Treatment and Prevention Strategies"

_viruses, 2023, doi:10.3390/v15122326_

Round 1

Reviewer 1 Report

Comments and Suggestions for Authors

This is an interesting review by Filippo Patrucco et.al. “Herpes Viruses Infection in Lung Transplantation: Diagnosis, Treatment and Prevention Strategies”. The review was well written with detailed Herpes virus infection in lung transplant patients. However, there are few places in the review in which the presentation must be improved as noted below.

Any other viral infection in lung transplant patients was observed?

Too many references were used in this review. Try to reduce the number

Lane 55: Is it “important” or “common” cause of morbidity?

Lane 59: which glycoproteins were responsible for entry? Is it membrane glycoproteins?

Lane 59: Remove “system”

Lane 67: It should be “After lung transplantation, the CMV can be acquired by following”

Lane 81: Should be “clinical manifestation on CMV infection”

Lane 91: Reference for selecting any two criteria?

Lane 101: should be “thrombocytopenia”

Lane 131: Elaborate figure 1 legend

Lane 140: why only “shell viral assay” is highlighted in this review eg. Figure 2?

  Lane 199: Elaborate figure 2 legend and point out the yellow and green spots and also red.

Lane 247: Should be :anti-IFN-gamma antibodies”. Is figure 3C is “CMV patient Specimen?”

Lane 250: explain “responder versus non responders”

 Lane 366: Is it “recipients systems”

Lane 513: Any cases of VZV infection in Lung transplant patients reported?

Lane 557: route of administration of acyclovir and valacyclovir

Lane 563: In which transplantation

Lane 597: 1-2 sentences of HHV-7 integration

Lane 598: remove “supposed to be the main”

Lane 607: explain “age15”

Comments on the Quality of English Language

no comments

Author Response

Reviewer #1

This is an interesting review by Filippo Patrucco et.al. “Herpes Viruses Infection in Lung Transplantation: Diagnosis, Treatment and Prevention Strategies”. The review was well written with detailed Herpes virus infection in lung transplant patients. However, there are few places in the review in which the presentation must be improved as noted below.

Any other viral infection in lung transplant patients was observed?

  1. R) Thank you for your comment. Many other viral infection have been observed in lung transplanted patients (i.e. influenza viruses, RSV, SARS-CoV-2) but in this review we preferred to focus our attention on Herpes viruses with the largest part represented by CMV and EBV and minor parts by other viruses. We did not report any information about HHV8 because data are only anecdotal, with a recent work by Tarsia et al (https://doi.org/10.1016/j.healun.2019.01.790). We would prefer to not introduce other viruses.

Too many references were used in this review. Try to reduce the number

Lane 55: Is it “important” or “common” cause of morbidity?

  1. R) CMV is both important and common, but we agree with you and we changed the text.

Lane 59: which glycoproteins were responsible for entry? Is it membrane glycoproteins?

  1. R) CMV surface glycoproteins gB and gH, which leads to activation of an NF-κB-dependent signal transduction pathway. We specified in the text.

Lane 59: Remove “system”

  1. R) Ok we removed

Lane 67: It should be “After lung transplantation, the CMV can be acquired by following”

  1. R) Ok we changed the text

Lane 81: Should be “clinical manifestation on CMV infection”

  1. R) Ok

Lane 91: Reference for selecting any two criteria?

  1. R) The reference is number 15 (Ljungman et al.); we added again the ref 15 before the list.

Lane 101: should be “thrombocytopenia”

  1. R) Ok

Lane 131: Elaborate figure 1 legend

  1. R) Ok we elaborated the legend by adding some details.

Lane 140: why only “shell viral assay” is highlighted in this review eg. Figure 2?

  1. R) Compared to traditional viral isolation, shell viral assay based on indirect immunofluorescence (IF) targeting CMV immediate early antigen (p72), reduces the time for viral detection. However, techniques based on viral isolation are now largely replaced by molecular assays.

Lane 199: Elaborate figure 2 legend and point out the yellow and green spots and also red.

Figure 2: Indirect immunofluorescence (IF) test. Nuclei of cells (human embrionary lung fibroblasts) infected by CMV appear green/ yellow while the non-infected nuclei appear red.

Lane 247: Should be :anti-IFN-gamma antibodies”. Is figure 3C is “CMV patient Specimen?”

  1. R) Yes we modified the text; Yes it’s patient’s specimen

Lane 250: explain “responder versus non responders”

  1. R) we added to the text the explanations (responder (recipients with a positive test) versus non responders (recipients with a negative test)

 Lane 366: Is it “recipients systems”

  1. R) Stems is changed with derives

Lane 513: Any cases of VZV infection in Lung transplant patients reported?

  1. R) A particular case of fatal septic shock after varicella zoster virus reinfection in lung transplanted patient was described by Lehingue et al. (1)The patient (62 years old) was VZV seropositive and was admitted to intensive care unit after lung transplantation complicated by graft dysfunction. After a few days a screening for VZV infection was conducted by quantitative blood polymerase chain reaction and analysis were positive with a viral load of 4.2x 104 copies/ml. Although the pharmacological therapies, the patient developed a multi-organ failure, never showed skin eruption, and died after nine days. The donor was seronegative for VZV, but VZV-specific molecular tests of donor lung biopsies and blood samples were performed and the results were positive. The fatal evolution is correlated with the disseminated VZV infection without skin eruption in the context of lung transplantation, as reported by Carby et al. (2).
  2. Lehingue S, Rambaud R, Guervilly C, Adda M, Forel JM, Cassir N, Zandotti C, Hraiech S, Papazian L. Fatal Septic Shock Triggered by Donor Transmitted Varicella Zoster Virus Reinfection 3 Days After Lung Transplantation. Transplantation. 2017 Dec;101(12):e351-e352.
  3. Carby M, Jones A, Burke M, Hall A, Banner N. Varicella infection after heart and lung transplantation: a single-center experience. J Heart Lung Transplant. 2007;26(4):399-402.

Lane 557: route of administration of acyclovir and valacyclovir

  1. R) Acyclovir (oral or IV administration), Valaciclovir (oral administration)

Lane 563: In which transplantation

  1. R) kidney, kidney-pancreas, liver, or heart transplants. We deleted “In one study 263 cases of SOT recipients were treated with oral valaciclovir for one year after transplantation and no cases of VZV infections were ob-563 served [126]. In these cases, prolonged antiviral prophylaxis is recommended for VZV 564 seropositive patients in chronic immunosuppression.”

Lane 597: 1-2 sentences of HHV-7 integration

  1. R) Integration of HHV-7 in human genome has not yet been proven, and it is a very specialized topic. However, as requested, we explored the topic more in depth:

“It establishes latency in CD4 lymphocytes [164]. HHV-7 integration into the human genome has been hypothesized [164]. The presence telomeric repeats sequences (TRS) in HHV-7 as in other Herpesviruses genomes suggests the possibility of homologous recombination between the telomeric region of human chromosomes and viral TRS facilitating the integration. However, at this moment, only HHV-6 integration into human chromosomes has been proven [164].”

Lane 598: remove “supposed to be the main”

  1. R) Corrected: “Saliva is the main transmission route in young age.”

Lane 607: explain “age15”

  1. R) “age15” is an error: this number was referring to a bibliographic entry about HHV-7 in our draft. The correct sentence is “HHV-7 primary infection is generally acquired in childhood within 5 years of age.”.

Reviewer 2 Report

Comments and Suggestions for Authors

This review paper provides a comprehensive overview of the epidemiology, diagnosis, and treatment of herpes virus infections in lung transplant patients. The authors highlight the high incidence of these infections, particularly CMV, and their significant impact on morbidity and mortality in this population. They also discuss the role of other herpes viruses, such as EBV, HSV, and Varicella-Zoster virus, in post-transplant complications. One of the strengths of this paper is the inclusion of new diagnostic tests and preventive strategies that have been developed in recent years. This provides valuable information for clinicians and researchers in the field of lung transplantation. However, it would have been helpful to have more specific and detailed information on these tests and strategies, including their accuracy and effectiveness. Additionally, while the review provides a good overview of the different herpes viruses and their impact on lung transplant patients, the section on treatment options could have been more comprehensive. Some of the treatments mentioned, such as antiviral therapy, are quite general and could have been further elaborated on. Overall, this review paper is a valuable resource for those interested in the management of herpes virus infections in lung transplant patients. It covers a wide range of topics and provides important insights on current diagnostic methods and treatment options. However, more detailed information on these topics would have enhanced the overall quality of the paper.

Author Response

This review paper provides a comprehensive overview of the epidemiology, diagnosis, and treatment of herpes virus infections in lung transplant patients. The authors highlight the high incidence of these infections, particularly CMV, and their significant impact on morbidity and mortality in this population. They also discuss the role of other herpes viruses, such as EBV, HSV, and Varicella-Zoster virus, in post-transplant complications. One of the strengths of this paper is the inclusion of new diagnostic tests and preventive strategies that have been developed in recent years. This provides valuable information for clinicians and researchers in the field of lung transplantation. However, it would have been helpful to have more specific and detailed information on these tests and strategies, including their accuracy and effectiveness. Additionally, while the review provides a good overview of the different herpes viruses and their impact on lung transplant patients, the section on treatment options could have been more comprehensive. Some of the treatments mentioned, such as antiviral therapy, are quite general and could have been further elaborated on. Overall, this review paper is a valuable resource for those interested in the management of herpes virus infections in lung transplant patients. It covers a wide range of topics and provides important insights on current diagnostic methods and treatment options. However, more detailed information on these topics would have enhanced the overall quality of the paper.

  1. R) Thank you for your appreciations; following your suggestions we:

- Viral cultures: we improved this section with more specifications on technical aspects.

- Improved details of Quantiferon tests: we added a very recent analysis conducted only on lung transplanted patients who received Quantiferon-CMV directed prophylaxis with interesting results.

Regarding the treatments of other herpes viruses (non CMV non EBV) literature details and recommendations on treatment are limited to small case series and strong evidences are lacking.

Reviewer 3 Report

Comments and Suggestions for Authors

The manuscript "Herpes viruses’ infection in lung transplantation: diagnosis, treatment and prevention strategies" represents a comprehensive literature review. Although lung transplantation is not a frequently used option, it is still an important treatment measure in certain patients having end-stage lung disease. Thus, the topic undoubtedly is timely and important, and it clearly corresponds to the scope of journal “Viruses”.

The article is well-written. It is characterised by clear design, logical structure and reasonably high level of English language, ensuring good scientific comprehensibility. The other strong points include the clear description of epidemiology and clinical manifestations for each virus as well as the focus on modern laboratory investigation methods.

There are only few suggestions for further improvements:

1) What are the main toxicities of ganciclovir and valganciclovir? What tests would you advise to detect and evaluate toxicity?

2) If possible, it would be nice to add morphological and immunohistochemical images, showing CMV infection (in contrast to allograft rejection) and HSV pneumonia. However, this is just a suggestion, not mandatory demand.

3) Please, check the formatting of references. It should be in accordance with the "Instructions for Authors".

Finally, I would like to thank the authors for their contribution. It was a pleasure and a true honour to review this manuscript.

Comments on the Quality of English Language

In general, the review is well-written and easy to understand. However, the text should be checked for minor misprints, e.g., Nocarida (line 128); embrionary (line 188); resources (line 147); sequence of words in a sentence (lines 189-190); gamma symbol (line 355) seems to show a misprint; words "freedom" (line 289) and "nevertheless" (line 476) are not appropriate in the given context. 

Author Response

The article is well-written. It is characterised by clear design, logical structure and reasonably high level of English language, ensuring good scientific comprehensibility. The other strong points include the clear description of epidemiology and clinical manifestations for each virus as well as the focus on modern laboratory investigation methods.

There are only few suggestions for further improvements:

1) What are the main toxicities of ganciclovir and valganciclovir? What tests would you advise to detect and evaluate toxicity?

  1. R) Thank you for the comment, we added to the text side effects related to ganciclovir and valganciclovir administrations.

2) If possible, it would be nice to add morphological and immunohistochemical images, showing CMV infection (in contrast to allograft rejection) and HSV pneumonia. However, this is just a suggestion, not mandatory demand.

  1. R) Thank you for the suggestion but we do not have explanatory images to add

3) Please, check the formatting of references. It should be in accordance with the "Instructions for Authors".

Finally, I would like to thank the authors for their contribution. It was a pleasure and a true honour to review this manuscript.

  1. R) Thank you for your appreciation

In general, the review is well-written and easy to understand. However, the text should be checked for minor misprints, e.g., Nocarida (line 128); embrionary (line 188); resources (line 147); sequence of words in a sentence (lines 189-190); gamma symbol (line 355) seems to show a misprint; words "freedom" (line 289) and "nevertheless" (line 476) are not appropriate in the given context. 

  1. R) we have made corrections that you suggested (gamma symbol was added by the ed.office); other grammar errors as well as minor misprints will be correct by editorial office.